# Climate Mitigation in the Swedish Single-Family Homes Industry and Potentials for LCA as Decision Support

Johanna Brismark [1,2], Tove Malmqvist [2,*] and Sara Borgström [3]

1    Plant An Idea AB, Storgatan 23C, 114 55 Stockholm, Sweden; johanna@plant.se
2    Department of Sustainable Development, Environmental Science and Engineering (SEED),
     KTH Royal Institute of Technology, 100 44 Stockholm, Sweden
3    Net Zero Buildings, Building Physics, Property and Buildings, WSP Sweden, Ullevigatan 19,
     411 40 Gothenburg, Sweden; sara.borgstrom@wsp.com
*    Correspondence: tovem@kth.se

**Abstract:** Decision support tools for incentivizing environmentally sound decisions in building design, such as LCA (life cycle assessment), have been highlighted as an essential feature for enhancing the realization of more sustainable buildings. Nevertheless, the use of LCA to support decisions in building design is still limited in practice. A better understanding of the social dynamics and detailed contexts of the decisions leading up to a final building design is therefore critical for better integration of LCA-based information in the decision-making processes. This paper reports a qualitative, semi-structured interview study of single-family home producers in Sweden and their decision-making in relation to climate mitigation, with a particular focus on embodied carbon mitigation. By studying a specific branch of the building and construction sector, a more in-depth record can be obtained of the particularities of implementation contexts and decision-making situations in which LCA may, or may not, have a role in driving climate mitigation. Four primary decision contexts in which LCA may have an influential role to drive embodied carbon reduction include: (1) the development of building systems, (2) development and offering of house models, (3) the selection of construction products for the building system as well as for the offer of add-on products to customers, and (4) the dialogues in the individual house-buyer projects. Decision-making that affects sustainable outcomes in this part of the sector is very much dependent on a supporting regulatory context. Over the years, using building LCA in early design stages, for optimization towards low-impact final buildings, has been a repeatedly promoted recommendation both in academia and practice. This study, however, reveals that such a conclusion is too simplistic. The different overarching decision contexts identified for this particular branch display the variety of needs for life cycle-based information.

**Keywords:** housing industry; life cycle assessment (LCA); decision-making; embodied carbon; interview study; buildings; climate impact

## 1. Introduction

Buildings are of significant concern regarding their large proportional environmental impacts in society; for example, climate impact and resource use [1,2]. While operational emissions still form the major challenge in most of the world, more recently, the need for reducing the large embodied carbon of buildings has become of increased concern for the industry and policy-makers [3–6]. Hence, with the raised interest in the embodied emissions of buildings, a novel interest in learning and using life cycle assessment (LCA) in building processes is also seen (ibid). Decision support tools to enable and incentivize environmentally sound decisions in building design have been a feature in the building and construction sectors since the end of the twentieth century [7,8]. Voluntary environmental certification tools have played a palpable role in shaping the sustainability agenda of the construction sectors in the last decades [8–10]. The LCA method was an integral

part of similar early tools, such as the German Legep, the Dutch Eco-quantum, and the Swedish EcoEffect [11], and later became embedded in certification tools such as LEED (Leadership in Energy and Environmental Design) and more recent tools such as DGNB (Deutsche Gesellschaft Nachhaltiges Bauen) and the method launched by the European Commission, Level(s).

Since the first building LCA tools in the early 1990s, this method has repeatedly been referred to as particularly useful for promoting building designs with low environmental impacts. However, its practical use for guiding building design decisions is still limited up to this date. It is commonly agreed upon that the use of LCA as decision support has the highest potential to promote a sustainable design if integrated at an early stage of the building design process, to work towards optimization of the specific and unique building at stake [12–17]. However, the primary use has instead been the contrary: to evaluate designs in later stages as part of building certification [17–19].

Much of the academic research on the LCA of buildings has focused on the development and implications of assessment methodology and its harmonization [20–22], rather than focusing on the context and decision situations in which these tools could be used in practice [23]. Scholars in the field have raised the need to find effective ways to integrate such quantitative decision support into the actual building processes, e.g., [15,16,24–27]. The challenges of efficient integration of LCA into building design processes, such as the contradictory and significant data needs in early stages and the high costs associated with the efficient arrival at reliable bills-of-quantities [13,14,17,28], have during recent years led to increased research activity focusing the integration of BIM and LCA, as well as other digitalized solutions for overcoming the afore-mentioned barriers [29–31]. Another strand of research includes those works that originate in numerical modeling disciplines, suggesting parametric design approaches based on multi-criteria decision-making. Life cycle climate impact is often one of the design optimization criteria, usually combined with construction costs or life cycle costing [32,33].

To summarize, science-based and technically focused work has dominated research on building LCA. Even though many of these studies also involve end-users, few tool and model development studies have been based on a profound understanding of the individual roles, actors, and decision situations in which the tools are to be used [23,34,35]. Thus, there is a need to better connect such technical knowledge to qualitative studies providing a more in-depth understanding of how and why environmentally critical decisions are taken in different types and parts of organizations belonging to the building and construction sectors [23,36,37]. Some examples do exist. For example, Moncaster [38] and Moncaster & Simmons [39] studied how environmentally critical decisions take place in building design processes for UK school projects. Reindl [40] and Willan et al. [41] highlight that in the field of energy efficiency studies in the property sector, building professionals are often under-studied as compared to households and occupants as actors, as seen, for example, in studies by Gram-Hanssen [42]. Reindl [40] used a middle-out perspective, which a few other scholars in similar studies have also used. For example, Willan et al. [43] delved into the dynamics of the multiple middle-actors between policy-makers and occupants affecting decisions for energy-efficient and low-carbon design in commercial properties with the help of this perspective. Leoto & Lizarralde [44] studied the complexity of integrated design processes and the need to enhance stakeholder interaction to provide more sustainable buildings. Related sociological studies in this field have been more focused on the institutionalizing of market standards, such as building sustainability certifications, in the shaping of the integrated decision-making concerning building sustainability, e.g., [9,45,46].

The study reported in this paper aims to reveal critical aspects for climate mitigation, and more specifically embodied carbon, in the building and construction sector by a better procedural understanding of the actual decision-making in building processes. Thus, the paper provides a contribution to the quite limited number of qualitative studies looking at decision-making for sustainability in the construction sector. This study zooms in on the single-family homes industry in Sweden. By studying a specific branch of the building



and construction sector, we have the opportunity to provide a more in-depth record of the particularities of decision-making situations in which LCA may or may not have a role in driving climate mitigation. Many of the actors representing the single-family housing industry in Sweden work with prefabricated concept buildings [47]. They deliver "above ground", meaning their contractors are responsible for the groundworks and foundation. The design processes thus differ much from those embracing architectural competitions or the design of specific, unique projects. However, increased prefabrication and building concept development are also seen in the multi-family buildings segment [48]. Thus, this study can elaborate findings of interest also for broader application than the single-family homes industry. Finally, the novelty of this study also lies in its focus on decisions concerning mitigation of the embodied carbon of buildings, allowing better reflection on relevant ways to use different types of life cycle-based quantitative information in various decision contexts.

## 2. Materials and Methods

The aim of the study was approached by conducting qualitative semi-structured interviews. Interviews of this type were deemed appropriate over, for example, surveys as they enable the capturing of experiences of the respondents and the underlying motivations for their decision-making, reasoning, and practices on a deeper level [49]. Five single-family home producers were chosen for the study. Three of the companies were already partners of the research project in which the study was conducted. To broaden the understanding of this part of the Swedish industry, two additional companies were selected. This was done in discussion with the sector organization leading the project, ensuring a relevant diversity of the studied sample regarding size, manufacturing type, and customer focus. However, the companies were also strategically selected because they needed to display some interest in sustainability to be of interest for the aim of the study. Therefore, they are not necessarily representative of all single-family home producers in Sweden, but provide an adequate selection of companies which have spent some reflection on their role in tackling climate change. Four out of the five companies are part of trade groups that cover various brands for single or multi-family homes and some also engage other parts of the value chain, such as sawmills, hardware stores, and project development. The Swedish single-family homes industry is almost entirely dominated by timber construction [47]. Some companies work with loose timber, but industrial production is otherwise dominant. Historically, manufacturers have mainly produced plan elements; however, lately, volumetric element production has increased, particularly when a few companies have entered the multi-family building market.

Between two to four respondents were selected and interviewed at each company to understand the different parts of their company's operations and to obtain a wider breadth of perspectives. A requirement was to find respondents that together could bring clarity to both the internal decision-making processes and strategic development work (for example, technical or sustainability managers) and those in specific development projects or orders (for example, sales managers). Dialogues were held with company representatives to arrive at a relevant selection of respondents. Finally, the sector organization was interviewed to provide additional insight into the industry's possibilities and barriers. Table 1 presents the respondents of the study and the companies they represented.

The interview guide was developed in iterations, based on discussions in the project group and a pilot interview with one representative of the companies. The interview guide aimed at capturing information about decision-making and potential use of LCA-based information both at the company level and at the individual building project level. Therefore, after initial open-ended questions about the company and role of the respondent, as well as perceptions concerning the sector's opportunities to mitigate climate change, the interview questions focused either one of these two levels, depending on the role of the interviewee in the company. A specific set of questions were also posed to the sector organization representatives on good examples of low-carbon decision-making, their

support to their companies, and experienced needs to accelerate climate change mitigation in the sector as a whole.

**Table 1.** Overview of studied companies and interviewed respondents.

| Company | Description | Role of Respondent | ID in the Paper |
|---|---|---|---|
| Producer 1 | Part of a larger trade group within the wood industry, industrial production. | Head of sustainability | P1 |
| | | Business manager branding | P2 |
| | | Business manager projects | P3 |
| Producer 2 | Part of a larger trade group with several brands, industrial production for the brand in question. | Head of sustainability | P4 |
| | | Head of the technology department | P5 |
| | | Division manager branding | P6 |
| Producer 3 | House supplier working with loose timber production. | Head of the technology department | P7 |
| | | Business developer, internal project manager energy | P8 |
| | | Head of building permit department | P9 |
| | | Project manager strategic projects | P10 |
| Producer 4 | Part of a larger (international) trade group, industrial production. | Head of the technology department | P11 |
| | | Development engineer R&D, energy, and climate | P12 |
| | | Product manager branding | P13 |
| Producer 5 | Part of a smaller trade group, industrial production. | Head of the technology department | P14 |
| | | Head of the architecture department | P15 |
| The sector organization | | Head of the technology department | P16 |
| | | Sustainability manager | P17 |

Thus, the interviews focused on understanding the decision-making processes at the different companies and the corresponding effects on the climate impact from detached houses. The interviews also explored specific prerequisites in the Swedish single-family homes industry and their implication on the companies' work with climate mitigation actions. Finally, the questions investigated how LCA-based information could support this work. To enable the generation of insights both into the formal and informal decision-making, a large part of the interviews were built around open questions, such as: "Could you tell me about a change/improvement done in recent years and how you ended up pursuing it?" or "Could you tell me about a house-buying project and the dialogue with the customer from the initial contact until the end of the project?" This format enabled interviews to open up for discussions based on the respondents' own experience and the possibility to adjust follow-up questions to dig deeper into each company's specific circumstances.

Information was sent in advance to the respondents to ensure a basic understanding of "LCA-based information". In addition, some respondents were asked to reflect in advance on a specific change or improvement they had been part of. Before their interview, all participating companies were asked to share any internal documentation of their process control or internal environmental management. However, in most cases, only limited information was shared. Instead, web pages were visited to generate a basic understanding of the companies' structure, type of offer, customer focus, and how they presented their sustainability efforts, policies, and targets.

All interviews were conducted over the digital conference tool Zoom, recorded, and then transcribed. The entirety of the transcripts was read in two steps. First, insights from the interviews were gathered in three mind maps, centered around key actors and their roles in decision-making as well as environmental work in this sector, with a focus on

embodied carbon mitigation and the use of LCA. Based on those insights and in discussions in the author group, four main decision contexts of interest emerged, which were then used to structure most of the rest of the analysis. The main point of these contexts was to both display the multitude of decisions with different characteristics of significance for the resulting environmental performance of the building, and to differentiate between decisions with high dependence of different key actors and employee roles. In the second step, more detailed quotes and insights were extracted and sorted under several identified themes and preliminary connected to the four decision contexts. In the writing process, further rearrangements and material reduction took place, and the empirical material was constantly revisited to ensure the validity of interpretations. Finally, the results chapter was sent out to the respondents to ensure correct interpretations.

## 3. Results

### 3.1. Environmental Considerations in the Single-Family Homes Industry

The interviews reveal that historically, environmental considerations in the single-family homes industry in Sweden have mainly dealt with improving the buildings' energy performance and, to some extent, reducing the presence of embedded hazardous substances through various product selection tools. In recent years, environmental certifications have begun to spread also to this part of the housing industry. Three of the studied companies work with the Nordic Ecolabel [50] and another one is considering it shortly. However, far from all projects are certified, as the certification process is often perceived as cost-driving. The Nordic Ecolabel for houses focuses on indoor environmental quality and energy use but is currently revised. There are requests from the respondents to cover embodied carbon in the certification. Apart from these aspects, the respondents also point out that industrial production could reduce material waste and facilitate waste sorting and recycling.

The respondents reflect that sustainability is now discussed at the management level and that sustainability concern is increasingly disseminated to additional parts of the companies. They perceive an increased awareness, even though there is a further need to spread knowledge within their respective organizations. A forum available at three of the studied companies at the trade group level is the so-called sustainability councils with representatives from large parts of the business. In addition, several companies have worked with sustainability targets in recent years, and four companies produce a yearly sustainability report. Sustainability targets cover, for example, energy use, fossil-free transport, reduced waste volumes, and better resource management. Some have also set sharp goals for climate neutrality (with varying scope), while some are still developing climate goals. Some respondents also raise the possibility of moving towards more circular business models. However, they see a significant challenge to including recycled material in their industrial production processes. Instead, respondents highlight the preparation for reuse of materials, elements, or entire modules.

### 3.2. Decision-Making Concerning Sustainability

In the following section, the decision-making in the studied companies is described as separated into four different decision contexts, all of importance for the final environmental performance of the buildings, but in different ways and with different implications:

- Development of building systems;
- Development and offer of house models;
- Product selection for the building system as well as the offer of add-on products for customers;
- Dialogues in the individual house-buyer projects.

In addition, most of the companies act as developers. This provides an opportunity to test alternative products and technical solutions in real projects. Some companies also carry out R&D projects. These activities sometimes influence the company's development in areas such as those listed above. One example with a strong climate connection, shared by P7, is to build a Zero $CO_2$-certified [51] villa to learn from. P12 describes a development project

with climate calculations and identification of climate reduction strategies concerning a preschool concept.

### 3.2.1. Development of Building Systems

The industrial single-family home producers work with locked technical platforms and automated lines to produce plan or volumetric elements in production facilities. As a result, there are technical limitations, such as wall height and material selection due to the production equipment. Any extensive changes regarding the building system therefore imply involvement from many parts of the company and careful investigations. Consequently, such changes are rarely implemented. Decisions regarding building systems are handled centrally and sometimes impact companies' brands and factories.

The respondents state two reasons for changes in the building systems. The most important are changes in legislation that place new demands on the performance of buildings. P5 details, for example, when they last made a change, in conjunction with stricter energy requirements in 2010. They redid their entire wall, even though they only started a new factory line 6 months earlier. They chose a wall solution that went beyond the stricter energy requirements as they saw a potential to position themselves in the market and for future-proofing.

The second reason concerns technical development; in this case, increased automation. P5 mentions their internal discussion of a loose-wool portal to increase automation in their factories. Even if automation drives the change, the climate issue will come into play because bio-based insulation is suitable in such a portal as it is "kind" to the equipment.

The respondents testify that evaluations of the product platforms are constantly ongoing, but, for example, P14 and P15 explain how they have always ended up not implementing any changes over the years. P7 recalls an extensive evaluation of their wall solution due to coming net-zero energy requirements. Fourteen completely different proposals were evaluated based on aspects such as the customer's total price, construction time, ease of construction for the contractors, and the designers' need for time and competence. Finally, they concluded that the wall they already had performed well.

Thus far, none of the studied companies had used LCA of any type that impacted decisions of developments of the building systems. However, one company incorporated climate calculations into their latest wall development project, but without influencing the final design of their ordinary wall concept.

For single-family homes in Sweden, the foundation and insulation normally account for a large proportion of the embodied carbon. This issue was touched upon in the interviews; however, respondents highlighted that it is an issue of lesser influence for these companies since it does not form part of the building systems they develop, and usually it is procured as a separate contract.

### 3.2.2. Development and Offer of House Models

The decisions made regarding house models are based, among other things, on sales statistics and market analyzes. They concern the development and decommissioning of different models and their design, such as floor plan, size, and areas. Apart from legal requirements, this is governed by the frames that their production technology set, cost efficiency, demand from the intended target group, and what is lacking in order to be able to present a broad offer for the clients. P13 explains, for example, that they aim to work with their standard components and technologies as much as possible, but that trends, such as large spans and open floor plans, imply that they may sometimes need to go outside their building system to remain attractive. The processes around the design of house models vary. P13 explains that they have a process in which various competencies evaluate sketches concerning aspects such as energy efficiency, fire safety, production technology, the Nordic Ecolabel, and accessibility. In P15's company, house models are developed by a group of architects with support from the technical department.

Today, parameters other than the environmental impact govern single-family home producers' development of house models. However, there are connections to climate considerations. P15 explains, for example, that they (especially for their low-cost models) work with simpler shapes and minimize low-quality floor space. P15 reasons that this approach is also material- and energy-efficient, even if economic concern rules the decisions. The same respondent adds that the basis for the pricing of their houses is on lengths and square meters of different building components, which entails that the increased use of materials directly affects the customer's cost. On the other hand, they still see that customers tend to be drawn to larger houses. It should also be emphasized that companies offer a wide range of models to meet the needs of all customers.

The companies' plans for how climate considerations should be included in the development process of house models were not brought up, except that P10 mentions that LCA should be used when designing new models, in addition to the cost. The significance of the individual house models for the resulting climate impact of the house-buyers projects differs much. The reason is that house models differ in how much they steer the customer's final choice. In some cases, the house models are just illustrative examples, while in other cases they become very decisive since there are fees for the customer if they want to make changes. Therefore, it might not be financially justifiable to, for example, change the size of the house.

### 3.2.3. Product Selection for the Structural Solution as Well as the Offer of Add-on Products for Customers

The environmental performance of single-family houses is highly dependent on the suppliers' material performance and product development. The respondents emphasize that competent suppliers can play a significant role in their product development and reduce production waste materials. The respondents believe that they can influence their suppliers to a certain extent, but it depends on how large a customer they are for their suppliers. However, several respondents raise the opportunity of joint dialogues with the suppliers via their industry organization.

Change of products or suppliers can be initiated for several reasons. For example, demand from customers and sellers, new products that suppliers present, contractors' experiences, input from the marketing department that keeps track of competitors, improvements concerning production technology, or experiences of complaints and problems with after-sales service. Decisions regarding products, both included in the building systems and for the offer of add-on products to customers, are usually handled by the companies' so-called product councils, in which a high representation of roles in the companies are present. Decisions are based on aspects such as technical performance, delivery options, guarantees, appearances, opinions from production or contractors, and price. For example, P14 says that they chose to start offering PVs on the initiative of their heat pump supplier, who developed a package offer, which made it easy for them to take in the product. P5 says that they have chosen to have a different type of insulation in their volumetric elements because they also produce such elements for apartment buildings. Since the fire requirements are stricter for such buildings, this steered the insulation choice also for their single-family home production. P7 also discusses insulation and how they wanted to switch to glass wool instead of rock wool, a change that could have reduced their climate impact, but their contractors opposed this because it risked worsening their working environment—the argument that became decisive in the end. P11 describes their decision process before offering an untreated facade material initiated by internal pressure and a perceived market need. After much research, decisive aspects included the Nordic Ecolabel, dimensional stability, and that it would be easy to handle in production. P11 points out that they excluded an alternative from New Zealand partly due to its long transport distance.

Respondents in three companies also describe how the Nordic Ecolabel now steers their product choice (P1, P4, P11, and P13). P1 explains that the Nordic Ecolabel requirements

are now the baseline of the product council, whereas earlier, it was primarily function that was discussed. Other aspects of the environmental performance important to companies are quality and service life since they want customers to talk well about them (P14, P11, and P8). In addition, P6 mentions origin as important, for example, prioritizing Swedish suppliers and products from local forestry to reduce transport. Two respondents in two different companies foresee that consideration of climate impact in their product decisions will be required in the future (P1 and P5). P7 and P10 state that it is already taken into account by the product council in their company. Three companies (mentioned by P5, P8, P10, and P12) have performed life cycle assessments of reference buildings to build up their understanding on hot-spot building components and materials, motivated by a view that it is a competitive advantage in competence building on embodied carbon.

Both P8 and P14 bring up that they need to introduce a new process or routine to obtain and manage product information on climate impact. Many respondents point out the importance of qualitative environmental supplier information to enable product choices. P5, P7 and P10 state that they increasingly request EPDs (Environmental Product Declarations) [52]. The knowledge level among many suppliers is, however, a problem. P8 and P10 believe that the situation may change rapidly as the pressure on suppliers to provide data increases. P4 notes that this may become a competitive factor among suppliers. Product substitution is thus the primary strategy brought up by the respondents to deal with climate mitigation. P12 mentions that it could be challenging to reduce the amount of material in their house.

Finally, respondents agree that some components are more challenging than others to change since they affect other production processes. Examples include kitchens, stairs, and windows. Several respondents also emphasize that they refrain from working with product changes that affect customers. Instead, in their climate improvement work, they focus on the products and materials that are the same for all customers to safeguard the customer's freedom of choice.

### 3.2.4. Dialogues in the Individual House-Buyer Projects

The decision situations described above concern each company's central work and the development of their offerings to the customer. Their building systems and house models set the frames for the houses to be built at each company, but additional choices are made in each house purchase. The companies commonly offer both locked-in, low-cost alternatives with few choices and house models open for more individual choices. Some also offer the possibility of building completely free architect-designed houses (within the frames of the building system). The potential customer choices may include interior and exterior product choices as well as the form and layout of the floor plan.

An agent or salesperson is the primary customer contact, but other functions may assist in customer dialogue. The respondents experience that the discussions with customers mainly deal with design and, to some extent, function (such as daylight considerations, type of surface materials, color choices, and floor plan layout), while they rarely raise technical and performance aspects. For example, P9 and P14 say they generally do not receive any questions about service life, and in the few cases that this occurs, it is primarily related to time spent on maintenance. P9, on the other hand, recalls that the number of questions about energy use, electricity costs, and PV's, has increased somewhat lately. Customers may also have culturally linked preferences; P9 mentions that the few non-timber houses they do sell are primarily in demand in southern Sweden, and P15 comments that facade materials such as brick and plaster are popular in the most southern part of Sweden.

In terms of environmental impact, P2 and P15 emphasize that the construction process is very complex for customers and environmental considerations therefore tends to be of a low order of priority. P2, P4, and P6 conclude it is primarily the cost that governs customers' choices, and P14 believes that most customers are only interested in knowing that the company complies with current regulations. P2 experiences that customers who buy more luxurious houses discuss the environmental and climate impact to a lesser extent.

They may, however, install PV since it is visible. On the other hand, low-price customers ask about the Nordic Ecolabel and the environment, which P2 believes is because they are younger and have a different awareness. Several respondents argue that the new generation of customers will be more aware and that it will be possible to discuss the climate impact of the houses with them.

Customers' direct opportunity to reduce the climate impact of their homes through the choices they make is valued differently among the respondents. Some see that their choices do not matter much and that it is the structural solution and the products used for the main structure that matter. Others believe that customers have a massive impact by primarily choosing how big a house they build and aspects such as the number of windows and the design.

P7-10 each talk about their company's plans to facilitate for the customers to make better climate choices by highlighting low-impact alternatives in their catalogues and working to more proactively communicate climate performance and improvement proposals. P7 mentions that they aim to present a preliminary climate impact for the house when they submit a tender, enabling the customer to understand better the climate implications of, for example, a larger house or more windows. P12 says they also want to improve their marketing of sustainable alternatives and reasons that one option would be to make them standard and let the customer instead opt-out if they want less sustainable alternatives. P3, P6, P9, and P14 still argue that communicating climate impact is difficult to laypersons. P9 suggests that one way to facilitate communication may be to offer "climate packages" in the same way some companies offer energy packages to help customers choose energy systems. Several respondents also point out that their sellers' knowledge of the climate issue is in many cases low, and they need support in presenting the issue, for example, through packaging. P1 highlights the Nordic Ecolabel as such a successful example, which nowadays is demanded by some customers after a work effort to develop the sellers' communication. At the same time, P9 and P10 believe that climate calculations, even if they are automated, are too complex to handle for salespeople and in their dialogues with customers. This is because a certain competence is still required to interpret the results and to be able to discuss potential improvements for the specific case.

3.2.5. Summary of Findings

Based on the narratives above, Figure 1 summarises the primary impacting factors for decisions taken in the realms of the four decision contexts. Based on the interviews, potential applications of LCA-based information to support low-carbon houses in the studied sector are also suggested, which is also further discussed in Section 4.2.

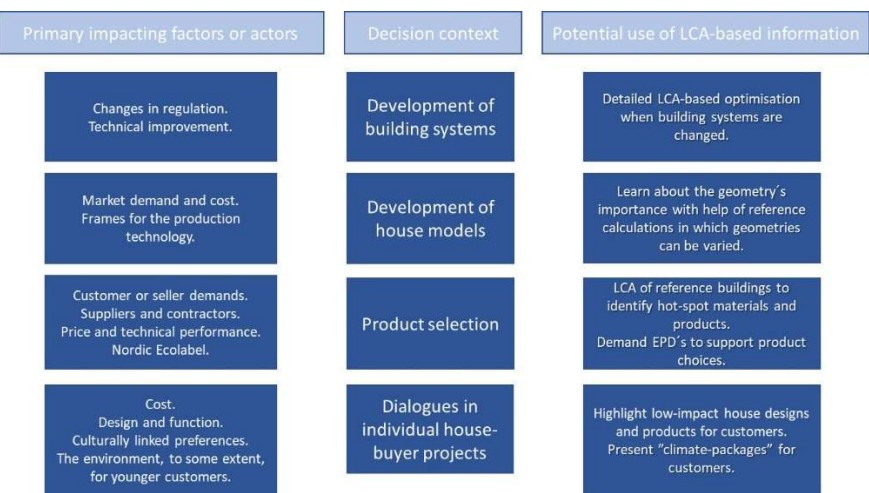

**Figure 1.** Summary of impacting factors and external actors in decision contexts of the single-family homes industry and potential use of LCA-based information to support low-carbon construction.

### 3.3. A New Regulatory Landscape

Thus far, Swedish building regulations have not dealt with buildings' climate impact, but as of 1 January 2022, a new regulation requiring a mandatory climate declaration of new buildings is in effect [53]. Several respondents foresee benefits from this legislation through raised awareness and motivation to address the climate issue by working with LCA. In a first step, they talk about internal learning on LCA methodology and the climate impact of the constituent materials of their product. Several companies have already begun calculations, more or less according to the regulation. There are examples where companies, thanks to these initial learning processes, have begun to pressure suppliers and demand EPDs, and in some cases, even question their climate data. P16 and P17 also describe how they at the industry organization see the effects of the forthcoming regulation by receiving requests for new development projects concerning alternative insulations and boards, and foundations in timber, due to companies' first analyzes. However, some respondents were not yet aware of the regulation, and P16 sees a need for guidance and support from, for example, their industry organization.

The single-family home producers have management advantages compared with other building developers because they know the construction products they use well. Many of the companies in the sector can probably incorporate climate impact calculations into their existing information systems, but an initial investment is required to enable this. However, private builders of single-family homes are exempted from the regulatory requirement on a mandatory climate declaration. Despite this, P6 reported they had already gone public with making climate declarations for all their houses, based on a desire to be prepared for further regulation, position themselves, and drive the industry forward. P16 believes that it is only a matter of time until more companies do the same. Some respondents believe they will implement climate improvement measures only if limit values are introduced in the regulation. However, the majority believe that they will use the knowledge they build up from completed climate calculations in their internal product development from the very beginning. Many also argue that single-family home producers, primarily constructing with timber, are well-placed in climate performance and will manage without excessive changes, even if limit values are introduced in the regulation.

## 4. Discussion

### 4.1. Critical Aspects for Climate Mitigation in the Single-Family Homes Industry

The interviews testify to the diversity of considerations that form the basis for the many individual decisions about changes to building systems, development of house models, product choices, and offers for customers, all of which ultimately impact the environmental performance of the houses. The respondents' reports show how the climate or environmental concerns have not been considered at all in critical decisions or maybe left behind for other aspects, but also how the issue has sometimes become significant or came as part of the bargain. The study reveals several examples of how the climate issue has begun to be considered as an additional aspect in decisions and ideas for better consideration in the future. For this to happen to a greater extent, the respondents primarily highlight increased customer demand and national regulation development. Other scholars have made similar observations, e.g., [24,54,55]. The respondents do not experience the former, apart from a certain tendency of increased awareness among younger customers. Instead, the respondents emphasize that regulation plays a central role in driving changes in this part of the industry. The introduction of limit values linked to the new regulation on mandatory climate declaration of new buildings, proposed by the Swedish authorities [56], is considered critical for climate concerns to become more heavily governed concerning future decisions in this industry.

The respondents see the most significant potential for reduced climate impact through structural solution changes and by selecting products with less climate impact. The first step in this work is to map the climate impact, as some have already done, to identify components and materials with a high climate impact (see e.g., [57]). The design and

product choices of the building system are also relevant because of the multiplying effect of industrial production dominating this part of the industry. In addition, the climate impact of the foundation is an important aspect to consider for the producers of single-family homes in the design of different house models and individual projects. However, this aspect has gained less attention due to the business model set up in this part of the industry.

The interviews reveal that these companies focus climate concerns on their structural solutions to prioritize climate considerations in areas that do not affect customers' possibilities when making their preferential choices. All the studied companies thus offer a wide range of house models to meet the needs of all potential customers. Åkerman et al. emphasize how home-buyers are very much in the hands of the professionals in the gradual shaping of their individual choices [58]. They conclude, when studying the barriers to penetration of innovative energy solutions in single-family home developments in Finland, that the professional mainstream planning, design, and routines are so institutionalized that even if the customers propose non-standard choices, it very seldom influences the final product (ibid). The present study's respondents similarly witness an intrinsic reluctance to face the climate aspect with the customers, thus resulting in conserving mainstream construction. In line with Åkerman et al. [58] it is too much to ask that the customers lead development by demanding low-carbon houses. However, the usual call for customer demand is raised among the respondents. This contradiction suggests an inertia regarding low-carbon development.

On the other hand, the interviews reveal that there exists companies willing to lead this development and which have ideas for enabling additional low-carbon choices for their customers. Similarly, Willan et al.'s "middle actor" respondents viewed themselves as having a role in guiding their clients towards improved energy efficiency [43]. It is nevertheless clear that single-family home producers have not yet formed a strategy to guide their customers to low-carbon choices. Moreover, respondents raised the pedagogical challenge since their customers are laymen and their sellers, who keep dialogue with the customers but rarely possess expert knowledge. In this context, the Nordic Ecolabel is highlighted as a successful strategy for communicating sustainability to the customer, not least because this does not require in-depth knowledge from the sellers.

Apart from a need for improving internal knowledge levels for the climate issue to be taken into account to a higher degree in decisions, the respondents in the study emphasize their construction material suppliers as important actors whose environmental competence is critical for supporting their work and aspirations. These actors have a role in providing comparable information regarding environmental footprints, driving product development, and assisting in waste reduction. The present study was performed as part of a project driven by the industry organization of the single-family homes industry in Sweden. Therefore, it serves as an example of how the organization and its member companies work with R&D projects to engine the sustainability work of the industry branch. As taken up in the interviews, a vital role of this organization can be dialogues with suppliers to raise joint forces demands concerning low-carbon products and better environmental information, such as EPDs. In particular, in a branch such as this one, much dominated by smaller actors, joining forces with the help of an industry organization may prove successful.

### 4.2. Implications for the Use of LCA-Based Information

One rationale for this study was to explore, in more detail, the decision contexts in which quantitative life cycle-based information could play a role in possibly supporting the mitigation of embodied carbon of buildings developed by the industry at stake. The interviews revealed several initial ideas and attempts to use such information, with further potential for development.

The recently installed regulation on mandatory climate declaration [53] seems to have already prompted the studied companies, probably leading to broad, in-house portfolios of climate-declared projects in the not-too-distant future. An advantage for this part of the industry is in-house project development, thus implying they may more easily work

with experience feedback of such calculations, feeding directly into potential tests of new alternatives in the next housing project. In addition, existing forums, such as the product councils in these companies, can facilitate changes and serve as role models for how other parts of the industry can work with integrated design processes.

Rigorous, comparative LCA of alternative solutions ought to have its natural place next to other in-depth investigations into the rare window of opportunity to change the building system. Moreover, industrial production enables establishing bills-of-resources with high precision to support such LCA applications. With the increasing engagement among the single-family home producers in the apartment building market, in which they use their industrial production and prefabrication of elements, such an application of LCA also becomes valid in their apartment building and other standardized building markets (such as preschools). However, a difference is that the rare window of opportunity primarily relates to the industrial production, whereas the platform development in, for example, the apartment building market ought to be seen as flexible scripts that can more consistently be open for development [59].

The traditional idea of building LCA as a tool entering early design stages for optimization towards a low-impact final building in individual projects can, therefore, at first sight, look less relevant for the single-family homes industry. However, an advantage of these industry segments is that it is not too hard to erect a bill-of-resources of a reference house model and then use it as a basis for investigating implications of alternatives of potential interest for a customer. Since most of the parameters are already "locked", such analyses would, on the contrary, imply fewer uncertainties than, for example, projects of architectural competitions and similar. The interviews revealed several potential usages of such a reference model: internal learning to identify hot-spot materials and components to be focused on for further improvements or to enlighten customers through climate footprints of house models and add-on products in their sales catalogues.

The previous section brought up the contradiction that the respondents ask for customer demands, but on the other hand, currently instead guide their customers into standard decisions, not considering climate impact from a life cycle perspective. Thus, it is argued that there is a need for these companies to pro-actively offer and market low-carbon alternatives, driving general raised awareness.

## 5. Conclusions

This study focused on Swedish single-family home producers and their decision-making in the perspective of sustainability, and more specifically concerning the topical issue of embodied carbon reduction in buildings. Four primary decision contexts, all of importance for the final environmental performance of the buildings, were identified: (1) the development of building systems, (2) the development and offer of individual house models, (3) product selection for the building systems as well as the offer of add-on products for customers, and (4) dialogues in of the individual house-buyer projects. The building system's development and product choices play an essential role in the resulting environmental performance and are decided centrally in the companies. However, in individual house-buyer projects, customer choices may impact geometries, the layout of floor plans, and the number of windows, which impact material intensities and consequently embodied carbon. In addition, customers' selection of add-on products may also differentiate the resulting environmental performance between individual projects. Some steps had been taken in the studied companies to increase internal knowledge concerning embodied carbon of their building systems and house models; however, communication with customers to guide them towards low-carbon choices seems to be the last outpost of climate mitigation work in these organizations, despite many of them being forerunners.

This study revealed that regulations, customer preferences, and product suppliers are critical actors/determinants that condition the possibilities for low-carbon construction in the Swedish single-family homes industry. The new regulation on a mandatory climate declaration of buildings seemed to provide new drivers for reducing embodied carbon

before its installment. However, the climate aspect is not yet on customers' agendas, resulting in space-inefficient layout plans and inferior form factors. This study nevertheless signals a slight tendency in customer preferences, implying that younger house-buyers are more concerned about the sustainability of the buildings they are buying.

The study also highlighted how an industry organization might have a vital role in driving sustainability concerns, particularly in industry segments dominated by smaller actors such as the single-family homes industry in Sweden. Apart from dialogues with suppliers, they may support the information and data management connected to climate calculations by disseminating show-case examples and developing standard formats, tools, and similar.

The study also aimed to explore implications for using LCA as decision support in this part of the construction industry. The study unfolds the decision-making and drivers in this part of the industry, which forms a basis for identifying ways to use LCA in fruitful manners. Firstly, in the context of the rare window of opportunity to change the structural solutions, rigorous comparative LCA of alternative solutions ought to have its natural place next to other in-depth investigations. This application of LCA is increasingly becoming relevant for the apartment building market since developers active in this market tend to increase their prefabrication and platform development.

In addition, the setup of a company-specific reference bill-of-resources can serve multiple purposes; internal learning to identify hot-spot materials, products, and components to focus on for further improvements or for illustrating the impact of product choices the customer takes. Since the windows of opportunity to make building system changes are rare, proactive and well-informed product substitution between such changes is critical for a faster motion towards net-zero emission targets. Such a reference model would facilitate the erection of aggregated climate declarations of different house model offerings to enlighten sellers and customers. Nevertheless, most of the studied companies can be seen as sustainability forerunners in this industry. Therefore, these forerunners can, and also seem interested in, playing a role in driving climate mitigation in the single-family homes industry. Then, they must act more proactively in customer dialogues and use LCA-based information to offer low-carbon alternatives as standard solutions and guide their customers towards low-carbon product selection. Thus, they could support general raising of awareness in this industry. However, most home producers cannot be expected to demand information from suppliers on climate performance, nor change their customer offers to more low-carbon. Therefore, regulatory intervention is necessary and examples of its effects were demonstrated in this study. In addition, the industry organization has a critical role in increasing the general knowledge level among industry members and supporting their climate practice.

Finally, the study revealed that the repeatedly promoted idea of complete building LCA as a tool in the early design stages for optimization towards a low-impact final building requires modification. The different overarching decision contexts identified for this particular branch display the variety of needs for life cycle-based information.

**Author Contributions:** Conceptualization, J.B., T.M. and S.B.; methodology, J.B. and T.M.; formal analysis, J.B. and T.M.; investigation, J.B.; writing—original draft preparation, J.B.; writing—review and editing, T.M. and S.B.; supervision, T.M.; project administration, S.B.; funding acquisition, T.M. and S.B. All authors have read and agreed to the published version of the manuscript.

**Funding:** This research was funded by the SWEDISH ENERGY AGENCY, grant numbers 49546-1 and 46881-1.

**Institutional Review Board Statement:** Ethical review and approval were waived for this study due to not handling personal information of sensitive character.

**Informed Consent Statement:** Informed consent was obtained from all subjects involved in the study.

**Data Availability Statement:** The data presented in this study are available on request from the corresponding author. The data are not publicly available due to privacy.

**Acknowledgments:** The authors gratefully acknowledge funding received from the Swedish Energy Agency and the authors would like to direct great appreciation to the interview participants and organizations who contributed to the study.

**Conflicts of Interest:** The authors declare no conflict of interest.

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
