# Peer review of "Climate Mitigation in the Swedish Single-Family Homes Industry and Potentials for LCA as Decision Support"

_buildings, doi:10.3390/buildings12050588_

Round 1
Reviewer 1 Report
In current study (Buildings- 1658219), the article presented: Climate mitigation in the Swedish single-family homes industry and potentials for LCA as decision support. The manuscript tried to study of single-family home producers in Sweden. The paper can be considered for publication with minor correction.
Abstract
The content of abstract is low, it’s better revised.
The Lines 11-14 are better reconstructed
Life cycle assessment (LCA) as decision support tools …. .
Line 16 change with, This paper investigated …
Lines 18-21, in the sentence used could, may, … it’s like suggestion. If it is your result or conclusion, its better reconsidered sentence structure.
Line 21, again use “could” and its better changed.
The sentence in line 27 is ambiguous
Introduction
The authors always used bindle references
Line 44, (LEGEP), its used for first time, and need put certain word with abbreviation, again repeated in line 45, 46,…
Line 78, what’s that 36 36 36???
What’s this (see e.g [36], [37,38].) it’s better use different techniques for writing about examples
Line 91 (The study reported in this paper aims…) It’s better to change with (this study aims…)
Line 96 and 21 repeated again (decision-making situations in which LCA may, or may not have a role in driving climate mitigation)
Methods
Line 109, the end of sentence is better mention, (semi-structured questions as interview research method)
It’s better the authors mention here why they consider this type of interview for this study, for example consider the advantages of this type of interview, also mention it in abstract, too.
Reviewer 2 Report
This paper reports a study of single-family home producers in Sweden, and their decision-making in relation to climate mitigation, with a 17 particular focus on embodied carbon mitigation from five companies. The results and discussion were supported by interviews. The following comments should be addressed in revising the paper.
- line 78, wrong citations of reference [36].
- line 116, as the authors claimed that the five selected companies were not necessarily representative of all single-family homes producers in Sweden, how can they ensure the objectivity and representativeness of the results and discussion.
- lines 135-145, provide more details about the interview which was the main research method of this paper. For example, the authors should explain how have they designed the questions in the interview, and list the main questions.
- Section 3, use of tables or graphs to summarize the answers by different respondents can be helpful to promote the readability of this section.
- as a scientific paper, the author should better highlight the scientific nature and novelty.
Author Response
Please, see the attachment.

Reviewer 3 Report
Dear Authors,
First of all, I would like to commend you on providing an interesting mansucript. Introduction of mandatory environmental impact calculations as well as company-provided "climate packages" might serve as inspiration for other countries and regions. Sadly, the quality of presentation should be improved in my opinion. Therefore I would like to suggest the following changes:
ENGLISH AND TERMINOLOGY
Overall, the linguistic quality of the manuscript is good. There are only some occasional minor errors, such as:
- Lines 45-46: LEED, DGNB and LCA are not "tools", but "methods".
- Line 47: "1990ies" instead of more common "1990s"
- Line 52: "... design process of buildings..." instead of ..."building design process..."
- Line 154: "zoom" instead of "Zoom". Also, it could be highlighted that this is an online conference tool for those not familiar with it.
- Line 206: "... homes producers..." instead of "... home producers..."
- Line 325 (and two other instances): Use of EPD abbreviation without explanation. 1) It should be mentioned that it stands for Environmental Product Declaration; 2) A reference to additional information would be beneficial for anyone not familiar with it.
The terminology in the manuscript requires improvements and clarification. Manuscript commonly uses words such as "many", "most of", "some" and "several" in combination with "interviewed emploees", "respondents", "interviewees", "companies", ... when describing results of the surveys. This use of (probably) synonyms and adjectives makes the text needlesly confusing and inaccurate in my opinion. It is often unclear 1) if the text refers to a number of companies or a number of survey participants; 2) what the particular adjective represents. It would be more accurate to provide specific information/numbers. For instance:
- The text in line 180 refers to "some" and text in lines 183-184 refers to "several" companies. Are both sentences referring to the same companies?
- Line 309 says that "Many companies..." Based on the listed survey participants (P1, P4, P11, and P13 at the end of the sentence) it seems that you are referring to two companies. That does not qualify as "many" in my opinion.
- Line 469: What does "... most of the companies studied..." mean?
METHODS AND RESULT PRESENTATION
These sections are rather vague in my opinion. I would like to suggest following modifications to description of methods:
- Adding more information on "interview guide" and "pilot interview" mentioned in lines 136-137 or at least its summary (or list of questions) might be beneficial for comprehensibility of results. It could be added for example as a separate appendix.
- Lines 125-127 indicate that a number of emploees in each surveyed company were interviewed and subsequently 2-4 emploees in each company were selected as respondents. It is unclear if there's any overlap between the initially interviewed people and later selected respondents.
- If possible, please add more information regarding creation of the "four decision contexts" introduced in the abstract and lines 191-197.
Description of results includes only plain text. This text contains a lot of information and might be hard to follow. I would like to suggest:
- Lines 155-156 mention "three mind maps". Adding these (or their derivates) could significantly increase clarity of presentation of the methods and results in my opinion.
- Adding charts or tables with summary of responses per company/respondent would greatly increase clarity of the result presentation in my opinion.
- The results mention that some of the companies have already done LCAs and identified hot spots that need further attention. For example line 450 mentions foundations. Would it be possible to list these (with benefits/negatives, challenges, number of responses, etc.) in a table, bulletpoint list, ...? It might be beneficial for readers in other regions who work with similar typologies and construction systems.
SPECIFIC COMMENTS
- Line 36 states that "... need for reducing ... embodied carbon...". That's true, but operational emissions (HVAC, etc.) are more pressing issue in most of the world. Therefore I recommend adding at least a brief comment on this fact.
- Text in lines 39-42 would benefit from additional references.
- Line 44: Please, specify/elaborate on what you mean by "not until later".
- Line 78 includes reference [34] two times and reference [36] eight times in a row. Please, remove the escessive ones.
- Lines 119-120 state that "The Swedish single-family homes industry is entirely dominated by timber construction." Please, specify what this means, because it contradicts text in lines 355-357 which states i.a. "... lightweight concrete houses are primarily in demand in southern Sweden...". Some national statistics (chart, simple table, etc.) or at least reference to them would be beneficial.
- Lines 164-168: A reference would increase clarity and credibility of the text.
- Line 186 mentions some significant challenges related to recycling (followed by statement that re-use is preferred). If possible, please expand this information.
- Lines 273-275: I would like to recommend adding more information on "...the respondents discuss how the customers' decisions... may affect the climate performance". Other parts of the manuscript mention for example higher awarness of younger people with regard to environment.
- Lines 391-393 state that respondents said that "climate calculations are... too complex to handle". If possible, add more information on this issue. Is it due to lack of education, time, will, ...?
Author Response
Please, see the attachment.

Reviewer 4 Report
The proposed topic is interesting because could show the significant role of LCA at the early design stage instead of the detailed stages, and mainly a way on how to involve all the stakeholders involved in the design stage. However, Two things are missing:
- An exhaustive explication of the adopted method. It is not clearly evident how the selection of interviewers was conducted; Please add a sentence before table 1. Material and method section requires a strong review.
- A clear description of the results. it is complicated to understand the results without a canvas diagram or another graphical tool that summarise and connect the issues identified during the interviews.
Author Response
Please, see the attachment.

Round 2
Reviewer 2 Report
I am happy that the authors have revised the manuscript according to the comments.